# The Efficacy of Radiotherapy without Surgery for External Auditory Canal Squamous Cell Carcinoma

**DOI:** 10.3390/jcm11195905

**Published:** 2022-10-06

**Authors:** Naoto Osu, Atsushi Musha, Hikaru Yumisaki, Kohei Okada, Nobuteru Kubo, Naoko Okano, Yukihiko Takayasu, Masato Shino, Osamu Nikkuni, Shota Ida, Hidemasa Kawamura, Kazuaki Chikamatsu, Tatsuya Ohno

**Affiliations:** 1Department of Radiation Oncology, Gunma University Graduate School of Medicine, 3-39-22, Showa-machi, Maebashi 371-8511, Japan; 2Gunma University Heavy Ion Medical Center, 3-39-22, Showa-machi, Maebashi 371-8511, Japan; 3Department of Otolaryngology-Head and Neck Surgery, Gunma University Graduate School of Medicine, Maebashi 371-8511, Japan

**Keywords:** radiotherapy, chemoradiotherapy, external auditory cancer, temporal cancer, squamous cell carcinoma, 3D-CRT

## Abstract

External auditory canal (EAC) cancer is a rare disease for which there are no adequate evidence-based treatment strategies. Radiotherapy is often used as the initial treatment to preserve the organ. This study aimed to elucidate the efficacy of radiotherapy for EAC squamous cell carcinoma (SCC). Patients with T1 disease were treated with radiotherapy alone. Patients with T2–4 disease were treated with chemoradiotherapy. The median follow-up period was 30.4 months. The 3-year local control (LC) rate for all patients was 51%, the disease-free survival (DFS) rate was 44%, and the overall survival (OS) rate was 73%. For T1–3 disease, the 3-year LC rate was 74%, DFS was 62%, and OS was 89%. However, for T4 disease, the 3-year LC rate was 17%, DFS was 17%, and OS was 50%. In a univariate analysis, only the T-category was a significant factor for LC and DFS (*p* = 0.006 and 0.02, respectively). All local recurrences were within the high-dose irradiated area. The results of this study suggest chemoradiotherapy can be an alternative to a combination of surgery and postoperative radiation for T1–3 SCC of the EAC. However, the efficacy of chemoradiotherapy in T4 cases was inadequate.

## 1. Introduction

External auditory canal (EAC) cancer is a rare disease with an incidence of only 1 per 1,000,000 in the general population [1]. There are no large-scale studies on the treatment outcomes for EAC cancer, and no adequate evidence-based treatment strategies.

Several previous studies concluded that surgery alone or radiotherapy alone is recommended for early-stage EAC cancer, whereas a combinatorial treatment with surgery and chemoradiotherapy is recommended for advanced EAC cancer [2,3,4,5,6]. However, subtotal or total temporal bone resection is highly invasive. In recent years, radical radiotherapy has often been performed as the initial treatment for advanced EAC cancer, from the viewpoint of organ preservation.

Most of the data on radiotherapy outcomes for EAC cancer are based on conventional two-dimensional conformal radiation therapy, and there is a dearth of data on computed tomography (CT)-based three-dimensional conformal radiotherapy (3D-CRT) or intensity-modulated radiotherapy (IMRT) [3,7,8]. Some studies also mention the importance of evaluating tumor progression [9,10]. There is therefore a strong need to review the efficacy of CT-based radiotherapy.

This study aimed to elucidate the efficacy of radiotherapy without surgery for squamous cell carcinoma (SCC) of the EAC through a retrospective analysis of the outcomes of 3D-CRT or IMRT for patients with intact SCC of the EAC.

## 2. Materials and Methods

### 2.1. Patients

Fifteen consecutive patients with SCC of the EAC who were treated using chemoradiotherapy or radiotherapy alone at Gunma University Hospital between 2001 and 2021 were retrospectively analyzed. Two patients were treated using radiotherapy alone and thirteen patients were treated using chemoradiotherapy. All patients were histologically diagnosed with SCC of the EAC. The Pittsburg staging system was used to determine the T-category [11]. Lymph node metastases (N category) and distant metastases (M category) were classified using the 8th UICC TNM staging system for cancers of the head and neck region. The characteristics of the patients and their treatments are summarized in Table 1. The age of the patients ranged from 45 to 90 years (median 70). Their performance status according to the Eastern Cooperative Oncology Group classification was 0–1. Two patients had T1 disease, four had T2 disease, three had T3 disease, and six had T4 disease. Patients with metastatic lymph nodes localized to cervical lymph areas were included. One patient was classified as N1 and two as N2b. All patients were classified as M0. 

### 2.2. Treatment

All patients registered in this study underwent external beam radiotherapy planned on a Xio (Elekta, Stockholm, Sweden), Pinnacle (Pinnacle X-ray Solutions, Suwanee, GA, USA), or Eclipse (Varian, Palo Alto, CA, USA) treatment planning system according to CT imaging (2 mm slice thickness). The gross tumor in the EAC and any metastatic lymph nodes were delineated as the gross tumor volume (GTV). Contrast-enhanced CT, magnetic resonance imaging (MRI), or positron emission tomography-computed tomography (PET-CT) was used as the reference imaging to delineate the GTV. The clinical target volume (CTV) had at least a 5 mm margin around the GTV and included the entire lymph area where any metastatic lymph nodes were. The planned target volume (PTV) had a 5 mm margin around the CTV. The leaf margin of the multileaf collimator from the PTV was 5 mm. When the target was close to organs at risk (OAR), the leaf margin was modified to reduce the dose to the OAR. No patients received prophylactic irradiation of lymph node areas that were clinically determined to be free of metastases. Irradiation was performed with four, six, or ten megavolt linear accelerators: Synergy (Elekta), Oncor (Siemens Healthineers, Erlangen, Germany), or Trilogy (Varian). The fractional dose was 2.0 or 3.0, and the total dose was 66–70 Gy in all 15 patients. Three-dimensional CRT was performed on seven patients, and IMRT was performed on eight patients. IMRT was used primarily when it was considered that other radiation modalities would result in adverse events linked to irradiation of an organ (e.g., inner ear or brain) close to the tumor site, or when lymph node metastases were present, resulting in a complex shape of radiation distribution. All patients were immobilized using thermoplastic shells (Karin, Tokyo, Japan).

Two patients with T1 disease were treated with radiotherapy alone. One patient with T4 disease was treated with radiotherapy and concomitant intra-arterial cisplatin (CDDP; 150 mg/m^2^, four times). Twelve patients with T2–T4 disease were treated with concurrent chemoradiotherapy. Of the twelve patients who received concurrent chemotherapy regimens, six received intravenous administration of CDDP (60–80 mg/m^2^, weekly, three times), two received intravenous administration of CDDP (6 mg/m^2^, daily, 20 times) and docetaxel (DTX; 10 mg/m^2^, weekly, four times), and four received intravenous administration of cetuximab (initial: 400 mg/m^2^, 2nd–9th: 250 mg/m^2^, weekly). The treatment and clinical data of each patient are summarized in Table 2.

### 2.3. Follow-Up

After the completion of radical radiotherapy, all patients were examined by radiation oncologists and otolaryngologists. The follow-up intervals and imaging modalities differed across the patients. Recurrence was diagnosed by CT, MRI, PET-CT, or tissue biopsy. Acute and late adverse events were evaluated using the Common Terminology Criteria for Adverse Events, version 4.0 [12].

### 2.4. Statistical Analysis

The time data for local control (LC), disease-free survival (DFS), and overall survival (OS) from the start of treatment were analyzed using the Kaplan–Meier method. Univariate analyses of clinical and treatment factors were performed using the log-rank test. Statistical significance was set at *p* ≤ 0.05. All statistical analyses were performed using Prism8 (GraphPad Software, San Diego, CA, USA).

## 3. Results

### 3.1. Treatment Outcomes

Among the 15 patients, the follow-up period ranged from 7.6 to 120.8 months (median of 30.4 months). Twelve of the fifteen patients were alive at the end of the designated observation period (Table 2), whereas three patients had died from the primary disease. Eight patients experienced recurrence: six had local recurrence, one had cervical lymph node recurrence, and one had concurrent local recurrence and cervical lymph node recurrence. Five of seven patients with local recurrence had T4 disease before treatment. All local recurrences were within the high-dose irradiated area, and no patient had distant metastasis. The 3-year LC rate for all patients was 51%, the DFS rate was 44%, and the OS rate was 73% (Figure 1). For patients with T1–3 disease, the 3-year LC rate was 74%, the DFS rate was 62%, and the OS rate was 89% (Figure 2). For patients with T4 disease, the 3-year LC rate was 17%, the DFS rate was 17%, and the OS rate was 50%. Representative images before and after treatment and the dose distribution for a patient with T3 disease are shown in Figure 3.

The results of the univariate analyses are shown in Table 3. T-category was the only factor showing a significant association with LC rate (T1–3: 74% vs. T4: 17%, *p* = 0.006) and DFS rate (T1–3: 62% vs. T4: 17%, *p* = 0.020). No factors showed a significant association with OS, although T-category and radiotherapy dose approached significance at *p* = 0.071, and larger patient numbers may well have revealed significant associations.

### 3.2. Toxicity

The adverse events are summarized in Table 4. Acute adverse events included four patients with G2 dermatitis, four with G3 dermatitis, one with G2 mucositis, one with G2 nausea, one with dryness in the mouth, one with a G2 taste disorder, four with G2 leukopenia, and one with G3 neutropenia. All acute adverse events improved with symptomatic treatment. The only late adverse event was a G2 hearing impairment in one patient. No patient had facial or trigeminal nerve paralysis.

## 4. Discussion

In the present study, we retrospectively analyzed the results of radiotherapy in 15 patients with intact SCC of the EAC. The 3-year LC rate for all patients was 51%, the DFS rate was 44%, and the OS rate was 73%. In the univariate analysis, T-category was found to be a significant prognostic factor for LC and DFS. The 3-year LC, DFS, and OS rates in T1–3 cases were 74%, 62%, and 89%, respectively. However, in T4 cases, the 3-year LC, DFS, and OS rates were 17%, 17%, and 50%, suggesting that the efficacy of chemoradiotherapy for T4 cases was inadequate. It is difficult to interpret these results in the light of previous studies because many of the previous reports are based on heterogeneous cases, i.e., including inner ear cancer, different histology, and different staging [13]. However, the studies that seem most appropriate for comparison as the current situation were summarized in Table 5. Most studies indicate that T4 cases showed a poor prognosis after treatment (Table 5). 

Takenaka et al. performed a meta-analysis of data from 274 patients who were treated with chemoradiotherapy (and surgery) for EAC cancer [14]. They concluded that the treatment outcomes of chemoradiotherapy without surgery in T3–4 cases were comparable with those of surgery with postoperative radiotherapy, and that T4 cases had a significantly poorer prognosis than T3 cases. These findings are consistent with the results of the present study, in which the T4 cases had a poorer prognosis than the T1–3 cases (Figure 3, Table 5). In the present study, the LC rate of T4 cases was low (17%), and all five recurrences in T4 cases occurred in the high-dose area within the irradiation field. This suggests that T4 cases are resistant to X-rays, and that the chemoradiotherapy for T4 cases showed insufficient local efficacy. To address this issue, dose-escalation or more intense modalities are needed. For example, carbon-ion radiotherapy is a promising modality with a strong antitumor effect on X-ray-resistant tumors [15]. Koto et al. reported the treatment outcomes of carbon-ion radiotherapy for T3–4 squamous cell carcinoma of the EAC and middle ear [16]. Of the 13 patients in their study, three had T3 tumors and ten had T4 tumors. The LC rates at 1 and 3 years were 72% and 54%, respectively, and the OS rates were 70% and 40%. Although most of the patients had T4 tumors, the LC rate of carbon-ion radiotherapy appeared to be favorable compared with other studies using X-ray-based treatment, including the present study (Table 5).

Several studies on treatment strategies for EAC cancer concluded that a combination treatment of surgery and postoperative radiation was the most curative for T2–3 disease [3,5,6,17,18]. However, adverse events from surgery are not uncommon [3,19]. In cases with middle ear cavity invasion, subtotal temporal resection is often performed, and the facial nerve and inner ear are resected together. Generally, facial paralysis or hearing impairment caused by surgery does not recover [20]. In the present study, patients with T2–3 disease who refused surgery or who were considered unable to tolerate the invasiveness of the surgery were treated with chemoradiotherapy without surgery. Only one patient with T2 disease experienced local recurrence, and no patients developed a grade 3 or worse later adverse event. Although the number of T2–3 cases in our study was too small to compare the two treatment strategies, considering the balance between efficacy and toxicity, chemoradiotherapy as a primary treatment may be a reasonable choice for T2–3 cases.

**Table 5 jcm-11-05905-t005:** Data summary of the cited references.

Author	Total Number	Tumor Site	*n*	Pathology	*n*	# T-Category or Stage	*n*	Treatment			Outcomes		
									Group	Time Point	LC (%)	DFS (%)	OS (%)
Hashi et al. [3]	20	EAC	-	SCC	-	T1 T2 T3	8 8 4	RT ± surgery	All T1 T2	5 year	-	-	59 100 38
Yin et al. [4]	95	EAC ME	67 28	SCC	-	stageⅠ stageⅡ stage Ⅲ stage Ⅳ	22 17 18 38	Surgery ± RT ± chemo	All stageⅠ stageⅡ stage Ⅲ stage Ⅳ	5 year	-	-	66.8 100 100 67.2 29.5
Ogawa et al. [5]	87	EAC ME	59 28	SCC	-			RT ± surgery	All	5 year	-	54	55
						T1	13		T1			83	83
						T2	37		T2			69	69
						T3	37		T3			28	28
Choi et al. [6]	32	EAC ME	31 1	SCC ACC Other	21 9 2	stageⅠ/Ⅱ stage Ⅲ/Ⅳ	12 20	RT ± chemo ± surgery	All stageⅠ/Ⅱ stage Ⅲ/Ⅳ	5 year	-	52 65.6 41.4	57 70.7 48
Pemberton et al. [8]	123	EAC ME	53 70	SCC	-	T1 T2–3	27 96	RT	All	5 year	56	45	40
^#^ Takenaka et al. [14]	274	EAC	-	SCC	-	T1 T2 T3 T4	59 41 46 128	RT ± chemo ± surgery	All	5 year	-	-	57
									T1				-
									T2				-
									T3				72.5
									T4				35.8
Kato et al. [16]	13	EAC ME	11 2	SCC	-	T3 T4	3 10	Carbon-ion RT	All	3 year	54	-	40
Katano et al. [17]	34	EAC	-	SCC ACC Other	31 1 2	stageⅠ/Ⅱ stage Ⅲ/Ⅳ	7 27	RT ± chemo ± surgery	All stageⅠ/Ⅱ stage Ⅲ/Ⅳ	5 year	-	-	55.2 85.7 45.6
Nagano et al. [18]	21	EAC	-	SCC	-			RT ± chemo ± surgery	All	2 year	71	-	62
						T2	1		T2		100		100
						T3	10		T3		90		75
						T4	10		T4		50		10
Present study	15	EAC	-	SCC	-	T1 T2 T3 T4	2 4 3 6	RT ± chemo	All T1–3 T4	3 year	51 74 17	44 62 17	73 89 50

#, classified by Pittsberg staging system or Stell’s staging system; ACC, adenoid cystic carcinoma; Chemo, chemotherapy; EAC, external auditory canal; ME, middle ear; RT, radiotherapy; SCC, squamous cell carcinoma.

Nagano et al. showed the treatment outcomes of a combination treatment of surgery and postoperative radiation for T2–4 SCC of the EAC. In their study, the 2-year LC rate was 100% in T2 cases, 90% in T3 cases, and 50% in T4 cases. The 2-year OS was 100% in T2 cases, 75% in T3 cases, and 10% in T4 cases (Table 5). In the present study, the 3-year LC rate for T2–4 cases was 51% and the OS rate was 77% (Appendix A). Considering the difference in the assessment year, the LC rate of chemoradiotherapy in the present study is inferior to that of a combination treatment of surgery and postoperative radiation, but the OS rate does not seem so different. In interpreting LC rate and OS rate, we have to take into account that several patients in the present study received salvage treatment after local recurrence.

The optimal regimen for chemotherapy remains unknown. In the present study, patients with T2–4 tumors were treated by radiotherapy combined with concurrent CDDP, CDDP and DTX, or cetuximab. In the group who underwent radiotherapy with CDDP, there were three cases of T2 disease, two cases of T3 disease, and three cases of T4 disease. The CDDP was administered weekly, and the total dose was at least 200 mg/m^2^. In the T2–3 cases, only one out of five patients experienced local recurrence. A previous study recommended a total CDDP dose of more than 200 mg/m^2^ for nasopharyngeal cancer [21]. The results of the present study suggest that the total CDDP dose should also be more than 200 mg/m^2^ for EAC cancer. Of the two patients who underwent radiotherapy with CDDP and DTX, one had T2 disease and one T4 disease, and both achieved CR. It should be noted that among the T4 cases, only the combination of radiotherapy with CDDP and DTX resulted in CR. However, it is important to interpret this result carefully because only two patients underwent this treatment. Radiotherapy with cetuximab was also performed in T3–4 cases. One patient with T3 disease achieved CR, and their follow-up was terminated after 5 years. Two patients with T4 disease experienced local recurrence at a relatively early period (4.4 and 7.7 months), and the treatment effect of radiotherapy with cetuximab appears to be insufficient for T4 disease. The lack of uniformity in chemotherapy regimens in the present study is maybe due to complex reasons, i.e., lack of an established chemotherapy regimen for SCC of the EAC, the possibility that the indication criteria for chemotherapy in our hospital have changed due to the long duration of patient collection and differences in patient renal function.

In the present study, acute adverse events improved quickly, while late adverse events did not improve. One patient with T3 disease (patient number 7 in Table 2) experienced G2 hearing impairment as a late adverse event (Table 4). Marks et al. showed that the mean dose to the cochlea should be less than 45 Gy to avoid hearing impairment [22]. In our case, the mean dose to the cochlea was 64 Gy because the tumor was very close to the inner ear, and high-dose irradiation to the cochlea was unavoidable, despite using an IMRT technique. Hearing impairment should be avoided as it is directly linked to worse quality of life, but tumor localization can make it difficult to avoid hearing impairment in some cases, especially in T3–4 cases with inner ear invasion. When treating such patients, clinicians need to fully explain the possibility of hearing impairment at the time of informed consent. Brain necrosis, bone necrosis, and soft tissue necrosis, which have all been reported in previous studies, did not occur in the present study [8,18].

One limitation in interpreting the results of the present study is its single-institution retrospective design with a small number of patients. It took 20 years from 2001 to 2021 to collect 15 cases in the present study. As a result of the rarity of EAC cancer, it takes time to collect cases. Therefore, treatment methods may change over time when the cases are analyzed retrospectively. It would be desirable to concentrate cases in a single institution or to conduct a multicenter study, however, we believe that this is the best study that can be conducted at a single institution. Another limitation is the variation in treatment methods in terms of beam delivery technique, radiation dose, and combined systemic therapy. Further prospective studies using unified radiotherapy and concurrent chemotherapy are warranted. 

The limitation in comparing the results of the present study with a previous study is that many of the previous reports are based on heterogeneous cases, i.e., including inner ear cancer, different histology, and different staging [13]. Data analysis based on homogeneous cases is necessary to correctly evaluate the outcome of radiotherapy for SCC of the EAC.

## 5. Conclusions

In conclusion, we retrospectively analyzed the treatment outcomes of radiotherapy without surgery in 15 patients with SCC of the EAC. The results of the present study suggest that chemoradiotherapy is inferior to that of a combination treatment of surgery and postoperative radiation with respect to local control, but it can be an alternative to a combination of surgery and postoperative radiation for T1–3 SCC of the EAC, considering the invasiveness of surgery. However, the efficacy of chemoradiotherapy in T4 cases is inadequate. Further research is warranted to elucidate the efficacy of treatments with more intensive antitumor effects for locally advanced EAC cancers.

## Figures and Tables

**Figure 1 jcm-11-05905-f001:**
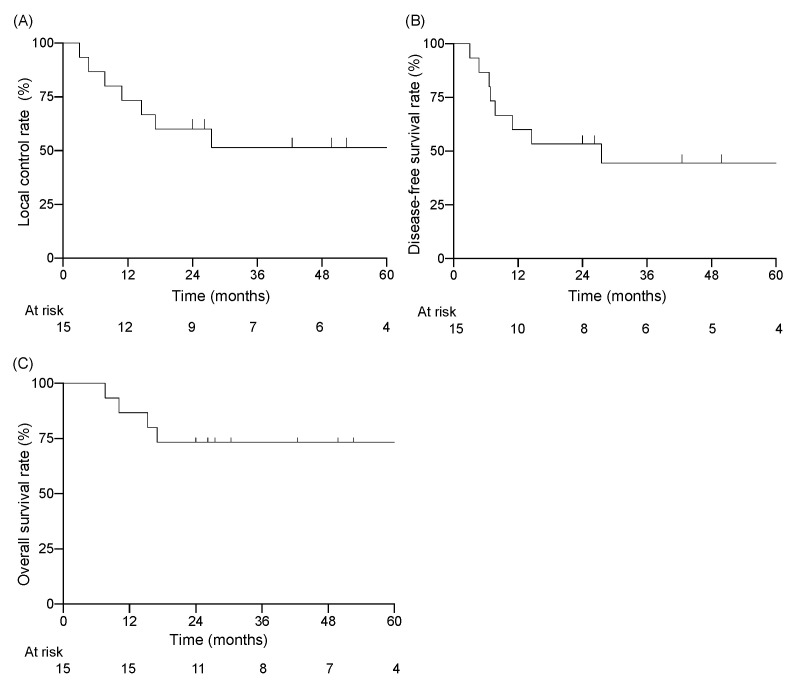
Kaplan–Meier curves for (**A**) local control rate, (**B**) disease-free survival rate, (**C**) overall survival rate.

**Figure 2 jcm-11-05905-f002:**
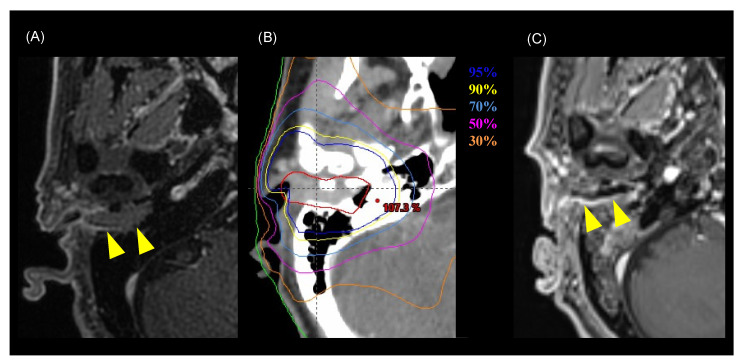
Representative images of a patient with T3 disease who was cured by chemoradiotherapy (Patient number 9 in Table 2). (**A**) MRI image before treatment. Yellow arrowheads indicate the tumor. (**B**) Dose distribution of the radiotherapy (intensity-modulated radiotherapy) on axial CT images. Highlighted are: 95% (blue), 90% (yellow), 70% (cyan), 50% (pink), 60% (magenta), 50% (purple), and 30% (orange) isodose curves (100% = 70 Gy). The red line shows the gross tumor volume. (**C**) MRI image six months after treatment. The tumor has shrunk and is not observable macroscopically. Yellow arrowheads indicate the tumor.

**Figure 3 jcm-11-05905-f003:**
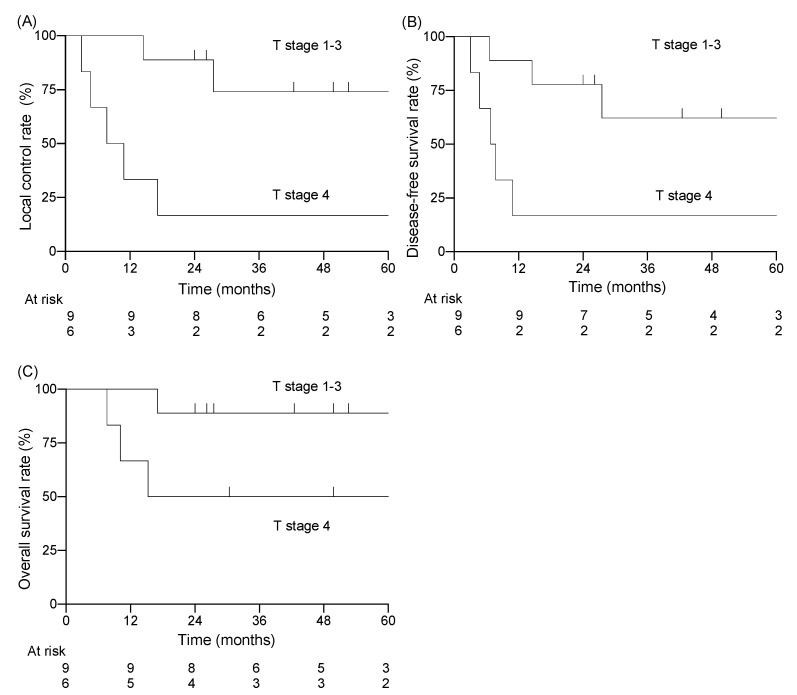
Kaplan–Meier curves stratified by T-category; T1–3 vs. T4 for (**A**) local control rate, (**B**) disease-free survival rate, (**C**) overall survival rate.

**Table 1 jcm-11-05905-t001:** Characteristics of the patients and their treatment.

Total	15
Age	
Range	45–90 years
Median	70 years
Sex	
Male	9
Female	6
PS	
0	9
1	6
T-category	
1	2
2	4
3	3
4	6
N-category	
0	12
1	1
2b	2
Total dose	
66 Gy	8
68 Gy	1
70 Gy	6
RT technique	
3D-CRT	7
IMRT	8
Concurent therapy	
CDDP	7
Cetuximab	4
CDDP and DTX	2
None	2

CDDP, cysplatin; DTX, docetaxel; IMRT, intensity-modulated radiation therapy; PS, performance status; RT, radiotherapy.

**Table 2 jcm-11-05905-t002:** Detailed patient information.

Patient Number	Age	PS	TNM Classification	RT Technique	Concurent Treatment	Gy/fr.	Reccurence	LC (M)	DFS (M)	OS (M)	Salvage Treatment	Status
1	90	0	T1N0M0	3D-CRT	-	44Gy/22fr. + 21Gy/7fr.	Local	14.5	14.5	17.0	-	DWD
2	67	0	T1N0M0	3D-CRT	-	70Gy/35fr.	None	42.5	42.5	42.5	-	AAW
3	50	0	T2N0M0	3D-CRT	CDDP+DTX	66Gy/33fr.	None	120.8	120.8	120.8	-	AAW
4	62	0	T2N0M0	3D-CRT	CDDP	66Gy/33fr.	None	49.8	49.8	49.8	-	AAW
5	64	0	T2N0M0	IMRT	CDDP	70Gy/35fr.	Local	27.5	27.5	27.5	-	AWD
6	66	1	T2N0M0	IMRT	CDDP	70Gy/35fr.	None	24.0	24.0	24.0	-	AAW
7	81	1	T3N0M0	IMRT	Cetuximab	66Gy/33fr.	None	68.1	68.1	68.1	-	AAW
8	74	0	T3N0M0	3D-CRT	CDDP	66Gy/33fr.	Lymph node	52.6	6.6	52.6	Salvage Surgery + Radiotherapy	AAW
9	70	1	T3N0M0	IMRT	CDDP	70Gy/35fr.	None	26.2	26.2	26.2	-	AAW
10	71	0	T4N0M0	3D-CRT	CDDP+DTX	66Gy/33fr.	None	65.7	65.7	65.7	-	AAW
11	72	1	T4N1M0	IMRT	Cetuximab	66Gy/33fr.	Local	7.7	7.7	10.1	None	DWD
12	74	1	T4N0M0	IMRT	CDDP	70Gy/35fr.	Local+Lymph node	17.1	6.8	30.4	Nivolumab	AWD
13	45	0	T4N0M0	3D-CRT	Cetuximab	70Gy/35fr.	Local	4.7	4.7	49.8	Salvage Surgery + Cyberknife	AAW
14	57	1	T4N2bM0	IMRT	CDDP (intra-arterial cysplatin)	66Gy/33fr.	Local	3.0	3.0	7.6	None	DWD
15	74	0	T4N2bM0	IMRT	Cetuximab	66Gy/33fr.	Local	10.9	10.9	15.3	None	DWD

PS, performance status; RT, radiotherapy; IMRT, intensity-modulated radiation therapy; CDDP, cisplatin; DTX, docetaxel; LC, local control; DFS, disease free survival; OS, overall survival; DWD, dead with disease; AAW, alive and well; AWD, alive with disease; 3D-CRT, three-dimensional conformal radiation therapy.

**Table 3 jcm-11-05905-t003:** Treatment outcome according to the clinical and treatment factors.

		LC		DFS		OS	
Factors	n	3y Rate	*p*	3y Rate	*p*	3y Rate	*p*
**age**							
≦70	8	56.3	0.59	60.0	0.27	87.5	0.25
≧71	7	42.9		28.6		57.1	
**Sex**							
female	9	50.0	0.86	50	0.86	83.3	0.43
male	6	53.3		55.3		66.7	
**T-category**							
T1–3	9	74.1	**0.006**	62.2	**0.020**	88.9	0.071
T4	6	16.7		16.7		50.0	
**PS**							
0	9	55.6	0.62	44.4	0.83	77.8	0.53
1	6	50		50		66.7	
**RT technique**							
3D-CRT	7	71.4	0.20	57.1	0.47	85.7	0.27
IMRT	8	25		25		62.5	
**Dose**							
<70	9	55.6	0.92	44.4	0.90	55.6	0.071
70	6	33.3		33.3		100	

LC; local control rate, DFS; disease free survival rate, OS; overal survival rate, PS; perfomance status, RT; radiotherapy.

**Table 4 jcm-11-05905-t004:** Numbers of adverse events.

	Grade 1	Grade 2	Grade 3	Grade 4–5
Acute Adverse Event				
Dermatitis radiation	6	4	4	0
Alopecia	4	0	-	-
Mucositis oral	1	0	0	0
Cheilitis	1	0	0	0
Nausea	0	1	0	0
Dry mouth	6	1	0	0
Dysgeusia	6	1	0	0
White blood cell decreased	0	4	0	0
Neutrophil count decreased	0	0	1	0
Late Adverse Event				
Hearing impaired	0	1	0	0

Adverse event names and grade are in accordance with CTCAE 4.0.

## Data Availability

The dataset generated and/or analyzed during the current study is not publicly available because it contains personal information, but anonymized data are available from the corresponding author on reasonable request.

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
