# Peer review of "The Efficacy of Radiotherapy without Surgery for External Auditory Canal Squamous Cell Carcinoma"

_jcm, 2022, doi:10.3390/jcm11195905_

Round 1

Reviewer 1 Report

the authors present a retrospective study on the use of chemoradiotherapy on external auditory canal squamous cell carcinoma. The cancer is rare, consequentely their cohort is sufficient, even because similar to the other "only RT" studies in literature. The aim to verify if a conservative treatement is comparable to a destructive one is always welcome, in order to grant the best possibile treatement to patients. The methods are strong anche the results interesting, even if, as the authors assert, insufficient to have definitive data or protocol; however, it's a first step. 

The only concern I would like to express are the folowing:

line 54: the authors say that their patients received radical radiotherapy, but 12 out of 15 received chemoradiotherapy. I think it would be better to specifiy this issue from the beginning, to be clearer and more correct.

line 54-56 : why this 5 patients did'nt underwent the "standard" protocol with surgery + chemio/radiotherapy? they refused surgery or it was a choice of the authors, based on first observation? If they have decided to treat the patient programming the present study it is no more retrospective but prospective

Reviewer 2 Report

Authors should be acknowledged for their effort in reviewing radiation therapy results in EAC SCCs.

This manuscript is interesting, however, shows some major limitations that should be addressed:

- Although the Authors showed a homogeneous case series (at least in terms of histology and site of origin), the articles selected from literature and brought to discussion are based mostly on heterogeneous cases. As correctly stated by the Authors, it is rather difficult to drive any conclusion from these articles, considering the heterogeneity regarding primary tumor site, histological type, and treatment, which are likely to bias any possible conclusion (see also Franz et al. Histological type homogeneity: a cornerstone in analyzing temporal bone malignancies data. European Archives of Oto-Rhino-Laryngology (2020) 277:3233–3234). For homogeneity’s sake, only studies based solely on EAC squamous cell carcinomas treated with radiation therapy and no salvage surgery should be considered.

- Author’s own series included most cases underwent salvage treatment due to a recurrence or persistence after primary radiation therapy ­± CT. In many centers, EAC carcinoma is often addressed with upfront surgery both in early and advanced cases. Authors should discuss more profusely their results in terms of recurrence rate in view of the existing surgical series (especially those limited to EAC squamous cell carcinomas).

- If the focus of this paper is radiation therapy, it must be acknowledged that there is a lack of homogeneity regarding concurrent treatment, as well as RT dose. I’d recommend trying to analyze separately the outcome of a subgroup as homogeneous as possible (i.e. radiation + CDDP).

- Conclusions are shortcut: in view of treatment heterogeneity and of the small sample size (even for a rare disease) it is hard to find evidence that “chemoradiotherapy is a reasonable first-line treatment for T1−3 SCC of the EAC”, as claimed by the Authors. I’d recommend a more descriptive approach in the conclusions, also disclosing the limitations of this study.

- The title is misleading: the presented data refer to a non-surgical chemoradiation approach rather than to radiation therapy itself.

Reviewer 3 Report

I feel this is very interesting original paper. The author discussed the latest radiation therapy for a rare cancer, cancer of the external auditory canal. The content is well understood. However, I am a little concerned about 1) the small number of cases and 2) the lack of uniformity in the means of radiotherapy and concomitant chemotherapy menu. I would like to say that the author should describe more about this point in the discussion part. Furthermore, T4cases points out the limitations of radiotherapy (chemoradiotherapy) in this study. Recently, the application of immune checkpoint inhibitors is expected, and I would appreciate it if the authors could provide us with their future thoughts on this T4 case. Thank you in advance.

Round 2

Reviewer 3 Report

Thank you so much for giving nice comment.